# Association between Charlson Comorbidity Index and positive blood cultures at a tertiary-care hospital in Indonesia

Patricia M. Tauran[1]*, Mansyur Arif[2,3], Direk Limmathurotsakul[4,5,6], Marlieke E. A. de Kraker[7,8], Alexander M. Aiken[9]

1 Dr Alberth H. Torey Hospital, Teluk Wondama, Indonesia, 2 Dr. Wahidin Sudirohusodo Hospital, Makassar, South Sulawesi, Indonesia, 3 Department of Clinical Pathology, Faculty of Medicine, Hasanuddin University, Makassar, South Sulawesi, Indonesia, 4 Mahidol Oxford Tropical Medicine Research Unit, Faculty of Tropical Medicine, Mahidol University, Bangkok, Thailand, 5 Nuffield Department of Medicine, Centre for Tropical Medicine and Global Health, University of Oxford, Oxford, United Kingdom, 6 Department of Tropical Hygiene, Faculty of Tropical Medicine, Mahidol University, Bangkok, Thailand, 7 Infection Control program, Geneva University Hospitals and Faculty of Medicine, Geneva, Switzerland, 8 WHO collaborating center, Geneva, Switzerland, 9 Infectious Disease Epidemiology Department, London School of Hygiene & Tropical Medicine, London, United Kigndom

* patricia.tauran1@alumni.lshtm.ac.uk

## Abstract

Blood culture (BC) tests are a scarce resource in low- and middle-income countries (LMICs); therefore, prioritization based on likelihood of positive results might be beneficial. We aimed to determine whether comorbidities in the Charlson Comorbidity Index (CCI) were associated with positive BC tests among patients with suspected hospital-acquired bacteremia. We analysed a retrospective cohort from health records at Dr. Wahidin Sudirohusodo Hospital, Makassar, Indonesia from 2015-2018. We applied multivariable logistic regression to identify associations between CCI score and the outcome of the first BC taken two calendar days after admission, adjusting for confounders. The primary analysis considered BCs positive for all pathogens. Of 3,875 adult patients who had their first BCs taken two calendar days after hospital admissions, 786 (20.3%) had their first BCs positive for any pathogen. Those included 371 patients who had their first BCs positive for *Staphylococcus aureus* (n=133; 35.9%), *Acinetobacter* spp. (n=84; 22.6%), *Klebsiella. pneumoniae* (n=58; 15.6%), *Escherichia coli* (n=63; 17.0%) and *Pseudomonas aeruginosa* (n=33; 8.9%). There was no association between increasing CCI score and positive BC (OR 1.01, 95%CI: 0.96-1.06, p=0.69) after adjustment for age, sex and other potential confounders. There was some indication that antibiotic use prior to BC test acted as an effect modifier between CCI score and positivity of BC (p=0.17). In this single-hospital study, no significant association was observed between CCI score and positive BC taken two calendar days after hospital admission. We suggest that other factors need to be investigated to guide BC testing, and that improving diagnostic and antibiotic stewardship, including increasing resources for BC testing prior to antibiotics among hospitalized patients are needed in LMICs.

**Data availability statement:** All relevant data are provided in the supporting information submitted with this paper (S1 File. CCI score and Blood culture dataset). This file includes the anonymized dataset used for the analysis in an Excel format.

**Funding:** This work was supported by NIHR–Wellcome Partnership for Global Health Research International Master's Fellowship (grant number 219651/Z/19/Z). This research was funded in in part by the Wellcome Trust (grant number 220211/Z/20/Z). For the purpose of Open Access, the author has applied a CC BY public copyright licence to any Author Accepted Manuscript version arising from this submission.

**Competing interests:** The authors have declared that no competing interests exist.

## Introduction

Blood culture (BC) tests are a crucial method to diagnose bloodstream infection (BSI) and guide antibiotic treatment. BC should be obtained if BSI is suspected and should precede antibiotic initiation for maximum sensitivity [1]. However, BC tests are rarely performed in low- and middle-income countries (LMICs), primarily due to financial challenges and limited institutional support [2,3]. Furthermore, the proportion of BC that are positive is often low, usually below 10% [4]. For those reasons, many clinicians in LIMCs are reluctant to perform BC [5]. This prevents BSI diagnosis and limits the availability of information for controlling antimicrobial resistance (AMR).

Given the importance of BSI diagnosis and the limitations of the BC test, a prioritization strategy could increase the chance of BSI detection in clinical practice within existing funding. We hypothesized that patients with more extensive comorbid conditions would be more vulnerable to BSI because most medical comorbidities cause some degree of immunosuppression and hence raise the probability of positive BC.

The Charlson Comorbidity Index (CCI) is a widely known scoring system for adult comorbidities [6,7], which generates a numerical score based on the presence or absence of 19 common comorbidities, with a further possible adjustment for increasing age. A major benefit of CCI is that it has been adapted for use with ICD-10 code which is available through regular hospital information systems and hence easily administrated [8]. Thus, use of the CCI might aid in identifying patients who should be prioritized for BC to reduce morbidity and mortality, while managing costs.

We aimed to investigate the CCI score and positive BC for any pathogen (primary analysis) and five priority pathogens (secondary analysis) from hospital acquired BSI while considering other relevant risk factors for positive BC (such as age, sex, ward, length of hospitalization (LOS) prior to BC test, admission type, direct ICU/HCU admission) as potential confounders and prior antibiotic treatment as effect modifier.

## Materials and methods

### Ethics Statement

Data for research purposes (data collection) were accessed between 18 March 2019 and 12 December 2019 with ethical approval from the Hasanuddin University Research Ethics Committee (Reference Number: UH18120948). The authors had no access to information that could identify individual participants during or after data collection. All data were fully anonymized before authors accessed them, and the Education and Research Department of Dr. Wahidin Sudirohusodo Hospital waived the requirement for informed consent.

For the data analysis conducted for this manuscript, the authors extended the local ethical approval from the Hasanuddin University Research Ethics Committee (Reference Number: UH20120948) and obtained additional ethical approval from the London School of Hygiene & Tropical Medicine MSc Research Ethics Committee (Reference Number: 29175).

## Study design and setting

This analysis used clinical, pharmacy and laboratory data from Dr Wahidin Sudirohusodo Hospital, Makassar, South Sulawesi, Indonesia from 2015 to 2018. Wahidin Hospital is a third-level public health facilities with 935 bed capacity. It receives referral patients from second-level health facilities for specialist services such as oncology and cardiology.

## Data source

Data used were routinely collected hospital electronic records which were extracted from the hospital information system. Further details of data collection are described elsewhere [9]. This study used data from all hospitalized adult patients with the first BC taken after two calendar days of admission over the study period. One patient could have multiple hospital admissions, and one admission could result in multiple BC tests. The first BC per patient for the full study period was selected for the main analysis. All BCs were included for a sensitivity analysis.

## Patient's comorbidities

Patient's CCI was the exposure of interest and derived from ICD10 codes of patients' clinical diagnoses based on **C**laims-based, **D**isease-specific refinements, **M**atching translation to ICD-10 and **F**lexibility to use (CDMF) CCI Coding Schemes [8]. We applied the relevant weight (1, 2, 3 or 6 points) for each comorbidity whilst considering the hierarchy rule [8], and used age-adjustment (1 point for 50–59 years, 2 points for 60–69 years, 3 points for 70–79 years and 4 points for ≥ 80 years) [7]. Hierarchy rules refer to the principle that when multiple related conditions are present in a patient's medical record, only the most severe condition contributes to the total CCI score. Example, if a patient has both cerebrovascular disease (1 point) and hemiplegia or paraplegia (2 points), only the higher score of 2 points for hemiplegia/paraplegia is included in the CCI total, since hemiplegia/paraplegia is a more severe consequence of cerebrovascular disease [8]. A unique CCI score was calculated per patient for each admission.

## Positive blood cultures

Positivity of BC taken two calendar days after admission (hospital-acquired BSI) was the primary outcome and was categorized as a binary variable (negative or positive for common contaminants [*Bacillus* spp. (not *B. anthracis*), coagulase-negative staphylococci, viridans group streptococci, *Aerococcus* spp. *and Micrococcus* spp [10]] vs. positive for any pathogen). For a secondary analysis, we examined positivity of BC taken two days after admission with priority pathogens. We excluded BC positive for other pathogens. We defined priority pathogens as the five most common pathogens in Wahidin Hospital that were included in the 2015 WHO Global AMR priority pathogen list. These were *Escherichia coli, Klebsiella pneumoniae, Pseudomonas aeruginosa*, *Acinetobacter* spp. and, *Staphylococcus aureus* [11]. Since our tertiary care center has a relatively high level of referrals from other healthcare centers, identification of community-onset organisms by the two-calendar day threshold would be biased, and these were therefore excluded.

Blood culture utilization rate (per 1000 patient-days) is the total number of blood cultures collected from 2015 to 2018 divided by the sum of all patient-days spent in the hospital during the same period and multiply by 1000. Patient-days refer to the total number of days that all patients spend in the hospital from 2015 to 2018.

## Covariates

Variables that are considered important confounders, based on a Directed Acyclic Graph (DAG), included age, sex, admission type, direct admission to intensive care unit (ICU) or high care unit (HCU), length of hospitalization (LOS) prior to BC test, ward of BC test and intravenous antibiotic prior to BC test (Fig 1). Admission type (emergency versus elective visit) and direct admission to ICU/HCU were considered as proxy variables of acute severity of illness.

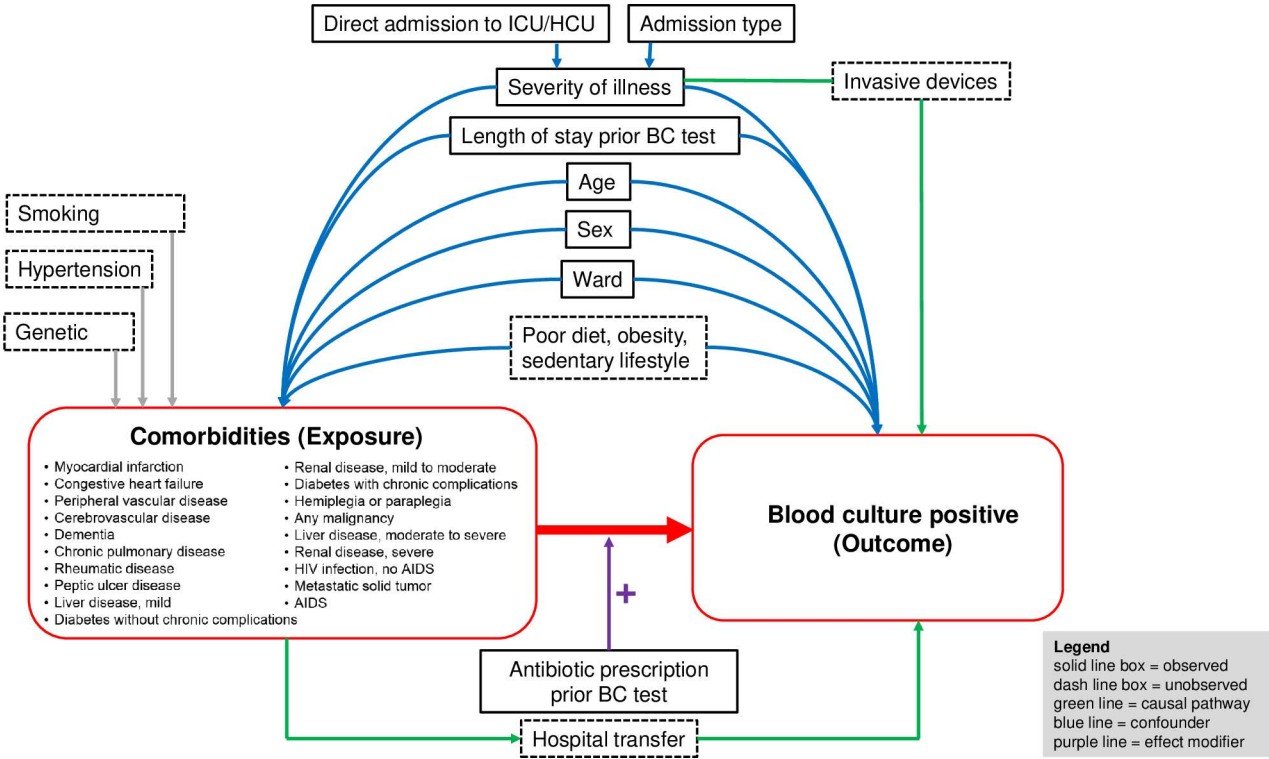

**Fig 1. Directed Acyclic Graph (DAG).** Footnote of Fig 1. Abbreviations: BC, blood culture; ICU, intensive care unit; HCU, high care unit; HIV, human immunodeficiency virus; AIDS, acquired immunodeficiency syndrome.

Admission type (binary) was defined as the department (emergency or outpatient clinic) that referred the patient to be hospitalized. LOS prior to BC test was handled as a continuous variable. Ward of BC test was categorized based on type (emergency, general, progressive or HCU and intensive or ICU). For multivariable analysis, wards were grouped into binary variable (emergency/general unit and progressive/intensive unit) due to small numbers in progressive wards. Intravenous antibiotic prior to BC test was a continuous variable and defined as the duration from the first date of any intravenous antibiotic prescription to the BC sampling date, regardless of whether the administration was continuous or intermittent and the number of antibiotics used. As an effect modifier, intravenous antibiotics prior to BC test were grouped into binary variables based on presence/absence of prior antibiotic treatment.

### Statistical analysis

Descriptive analysis was applied, using proportions, median and interquartile range, or mean and 95% confidence interval, whenever appropriate. Multivariable logistic regression analysis was performed to identify the association between CCI score (continuous independent variable) and BC positivity (binary dependent variable) for both the primary (first BC positive for all pathogens) and secondary (first BC positive for five priority pathogens) outcomes. For sensitivity analysis, all repeated BC tests were used as the outcome while considering random effects to take account of the within-cluster correlation variation as one patient could have several admissions or several BC tests during one admission. For patients with missing data of antibiotic dates, we assumed that no antibiotics were given when the BC was taken.

Likelihood ratio tests (LRT) were performed to compare nested models with CCI as categorical variable (CCI score of 0, 1, > 1) with the complex model including CCI as a continuous variable. "Use of antibiotics prior to BC test" was

hypothesized as a potential effect modifier because the impact of CCI on BC could be different depending on the presence/absence of prior antibiotic treatment, which was tested with LRT as well. Effect modifier was evaluated for any pathogens using the first BC taken after two calendar days of admission.

## Results

### Description of population and samples

There were 61,284 adult patients (total 98,102 admissions) hospitalized during four years of study with 4,562 (7.4%) patients having at least one BC test performed (total 5,704 BC). The total patient-days were 892,149, giving a BC utilization rate of 6.4 BC per 1,000 patients-days. Of those 4,562 patients, there were 3,875 (84.9%) patients with their first BC taken two calendar days after admission (total 4,797 BC) (Fig 2). These 3,875 patients were included in the study analysis.

Of the 3,875 patients, 460 (11.9%) had BC taken prior to antibiotic administration. Patients' characteristics are presented in Table 1. Minimal missing data was found for hospital discharge date (1/3,875; 0.03%) and antibiotic date (132/3,875; 3.4%). For the single patient with a missing discharge date, length of stay was imputed as one day based on identical admission, antibiotic start dates and antibiotic end dates. For 132 cases with missing data of antibiotic dates, we assumed that no antibiotics were given when the BC was taken.

### Blood culture tests

Out of 3,875 patients with their first BC taken two calendar days after hospital admission, 786 (20.3%) had their first BC positive for any pathogen, 2,705 (69.8%) negative or no growth, and 384 (9.9%) positives for common contaminants (Table 1). Within these 786 patients, 371 patients (47.2%) had their first BC positive for priority pathogens; *S. aureus*

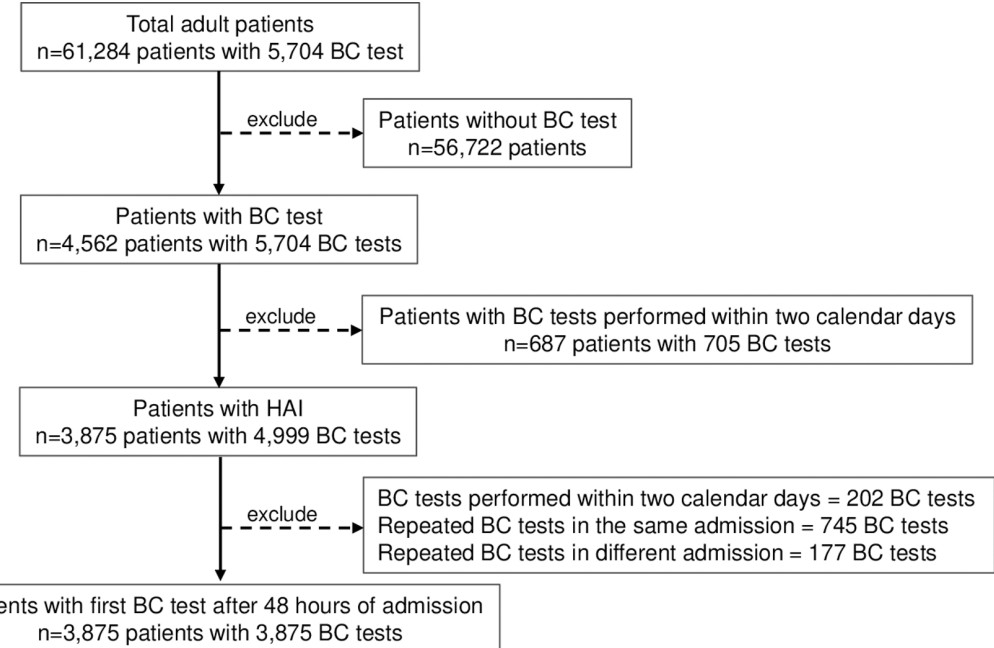

**Fig 2. Study flowchart. Footnote of Fig 2. Abbreviations: BC, blood culture.**

**Table 1. Characteristics of adult patients who had the first blood culture taken two calendar days after hospital admission.**

| Characteristics | Total patients N=3,875 patients (%) | BC negative N=3,089 patients (%) | BC positive for any pathogens N=786 patients (%) | BC positive for priority pathogens N=371 patients* (%) |
|---|---|---|---|---|
| Age on admission (years) | | | | |
| Median (IQR and range) | 51,1 (38.3-60.9, 18-97.4) | 51.1 (38.1-61.3, 18-97.4) | 51.0 (39.7-60.1, 18.2-94.9) | 51.7 (42.5-60.7, 18.2-87.7) |
| Sex | | | | |
| Female | 1,645 (42.5) | 1,307 (42.3) | 338 (43.0) | 180 (48.5) |
| Male | 2,230 (57.5) | 1,782 (57.7) | 448 (57.0) | 191 (51.5) |
| Admitted from | | | | |
| Outpatient department | 351 (9.1) | 250 (8.1) | 101 (12.8) | 49 (13.2) |
| Emergency department | 3,524 (90.9) | 2,839 (91.9) | 685 (87.2) | 322 (86.8) |
| Direct ICU/HCU admission | 607 (15.7) | 423 (13.7) | 184 (23.4) | 50 (13.5) |
| LOS prior to BC test (days) | | | | |
| Median (IQR and range) | 7 (4-14, 3-87) | 7 (4-13, 3-87) | 9 (5-17, 3-87) | 9 (5-15, 3-72) |
| Ward of BC test | | | | |
| Emergency | 23 (0.6) | 14 (0.4) | 9 (1.1) | 4 (1.1) |
| General | 3,028 (78.1) | 2,518 (81.5) | 510 (64.9) | 301 (81.1) |
| Progressive (HCU) | 299 (7.7) | 258 (8.4) | 41 (5.2) | 15 (4.0) |
| Intensive (ICU) | 525 (13.6) | 299 (9.7) | 226 (28.8) | 51 (13.8) |
| Antibiotic prior to BC test (days)** | | | | |
| Median (IQR and range) | 6 (4–11, 2–85); n=3,415 | 6 (4–11, 2–85); n=2,742 | 7 (4–14, 2–70); n=673 | 7 (3–14, 2–56); n=282 |

Abbreviations: BC, blood culture; IQR, interquartile range; ICU, intensive care unit; HCU, high care unit; LOS, length of stay.

*371 patients are part of the 786 patients.

**among patients who had antibiotics prior to BC

(n=133; 35.9%), *Acinetobacter spp.* (n=84; 22.6%), *K. pneumoniae* (n=58; 15.6%), *E. coli* (n=63; 17.0%) and *P. aeruginosa* (n=33; 8.9%).

## Patient's comorbidities

Overall, the median (IQR, range) of CCI score in all patients, patients with positive BCs, and patients with negative BCs were 2 (1–3, 0–11), 2 (0–4, 0–11), 2 (1–3, 0–11), respectively. Total CCI score and proportion of total CCI score, and BC positivity was showed in Figs 3 and 4.

The most common comorbidities observed in 3,875 patients were any malignancy (n=494; 12.8%), severe renal disease (372; 9.6%), congestive heart failure (361; 9.3%), diabetes with complications (324; 8.4%), diabetes without chronic complications (247; 6.4%), and cerebrovascular disease (229; 5.9%). The most common comorbidities observed amongst patients who had positive BC tests were severe renal disease (122; 15.5%), any malignancy (n=87; 11.1%), diabetes with complications (66; 8.4%), congestive heart failure (65; 8.3%), cerebrovascular disease (59; 7.5%) and diabetes without chronic complications (46; 5.9%).

## Association between CCI and positive BC for any pathogens (primary analysis)

In our main analysis using any positive BC as the outcome, we found that, there was no evidence that CCI score was associated with positive BC (adjusted odds ratio [aOR] 1.01, 95%CI: 0.96-1.06 p=0.69) (Table 2). Result of sensitivity analysis also support this finding.

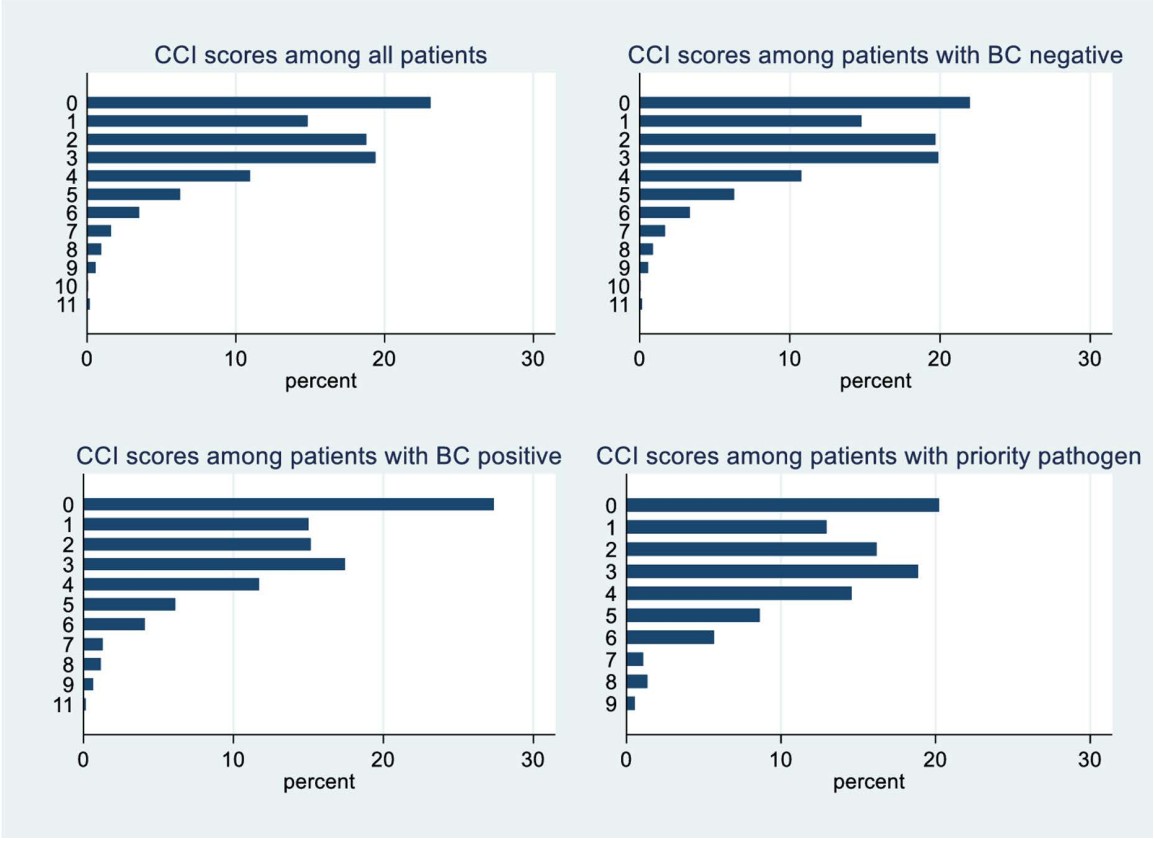

**Fig 3. CCI scores among all patients, and patients with negative, positive, and priority pathogen blood cultures.**

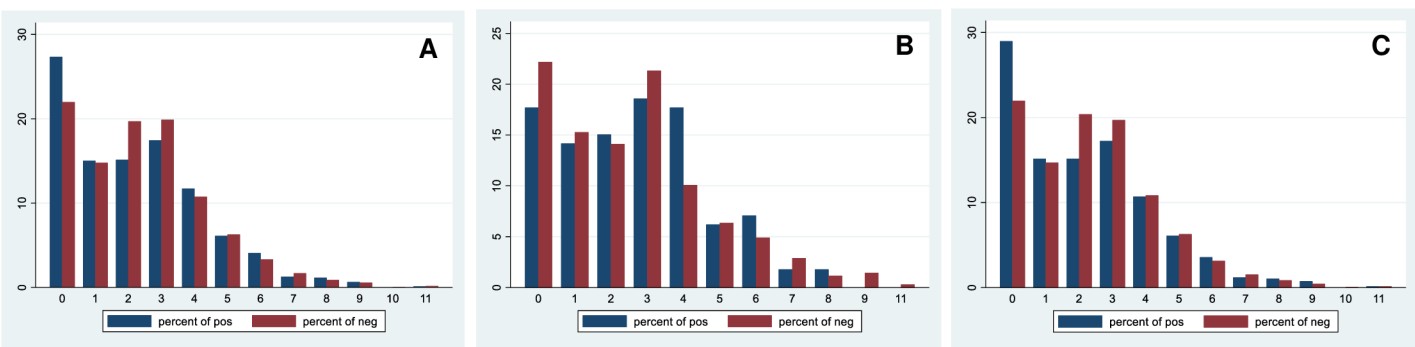

**Fig 4. A. Proportion of total CCI score and BC positivity in patients with first BC taken two calendar days after admission.** B. Proportion of total CCI score and BC positivity before antibiotic treatment. C. P**roportion of total CCI score and BC positivity after antibiotic treatment. Abbreviations: CCI, Charlson comorbidity index; BC, blood culture.**

Using CCI score as a categorical variable (CCI score of 0, 1, > 1) gave a better model fit (LRT, p = 0.069), and showed some indication that higher scores of CCI were less likely to have positive BC (aOR 0.84, 95%CI:0.63-1.11 and aOR 0.79, 95%CI: 0.62-1.01, p = 0.18) compared to CCI of 0, although confidence intervals included the null value.

**Table 2. Association between Charlson Comorbidity Index and blood culture positivity in patients with first BC taken two calendar days after hospital admission.**

| Patient's comorbidities (N = 3,875 patients) | BC test | BC positive for any pathogens | Adjusted OR* (95% CI) | p value | BC positive for priority pathogens[±] | Adjusted OR* (95% CI) | p value |
|---|---|---|---|---|---|---|---|
| Total CCI score (1–11, continuous) | | | | | | | |
| Main analysis[±] | 3,875 | 786 | 1.01 (0.96-1.06) | 0.69 | 371 | 1.08 (1.01-1.15) | 0.02 |
| Sensitivity analysis[#] | 4,797 | 1,055 | 1.03 (0.97-1.09) | 0.35 | 494 | 1.09 (1.01-1.19) | 0.04 |
| CCI score (categorical) | | | | | | | |
| 0 | 894 | 215 | 1 | 0.18 | 75 | 1 | 0.51 |
| 1 | 574 | 118 | 0.84 (0.63-1.11) | | 48 | 0.98 (0.66-1.46) | |
| 2-11 | 2,407 | 453 | 0.79 (0.62-1.01) | | 248 | 1.16 (0.83-1.61) | |

Abbreviations: BC, blood culture; OR, odds ratio; CCI, Charlson comorbidity index; LRT, likelihood ratio test.

*Adjusted with age, sex, admission type, direct admission to ICU/HCU, LOS prior to blood culture test, and ward of BC test. [±]Main analysis was calculated using multiple logistic regression.

[#]Sensitivity analysis was calculated using all BC test and random effect model. [±]*S. aureus, E. coli, K. pneumoniae, P. aeruginosa* and *Acinetobacter* spp..

## Association between CCI and BC positive for five priority pathogens (secondary analysis)

We considered an alternative outcome of a positive BC result with any one of five priority pathogens (Table 2). Using this outcome, there was evidence that increasing CCI was associated with BC positivity (aOR 1.08, 95%CI: 1.01-1.15, p = 0.02), which was supported by sensitivity analysis aOR 1.09 95%CI: 1.01-1.19 p = 0.04). Using CCI score as categorical variable (CCI scores of 0, 1, > 1) only showed an increased risk for CCI > 1 with wide confidence intervals (aOR 0.98, 95%CI: 0.66 -1.46 and aOR 1.16, 95%CI: 0.83 - 1.61, p = 0.51).

## Effect modifier

There was some indication that antibiotic prior to BC test acted as an effect modifier between CCI score (CCI score of 0, 1, > 1) and positive BC for any pathogens (Table 3) but this was not significant (p = 0.17). This non-significance may have

**Table 3. Multivariable models with and without effect modification*.**

| No Interaction | | | With Interaction** | | |
|---|---|---|---|---|---|
| Patient's comorbidities | OR* (95%CI) | p-value | Patient's comorbidities | OR* (95%CI) | p-value |
| CCI score (categorical) | | | No antibiotic prior to BC test | | |
| 0 | 1 | 0.18 | CCI score (categorical) | | |
| 1 | 0.84 (0.63-1.11) | | 0 | 1 | <0.001 |
| 2-11 | 0.79 (0.62-1.01) | | 1 | 1.16 (0.54-2.49) | |
| Antibiotic prior to BC test | | | 2-11 | 1.28 (0.72-2.29) | |
| No | 1 | <0.001 | Had antibiotics prior to BC test | | |
| Yes | 0.55 (0.43-0.70) | | CCI score (categorical) | | |
| | | | 0 | 0.84 (0.49-1.43) | |
| | | | 1 | 0.66 (0.38-1.16) | |
| | | | 2-11 | 0.61 (0.36-1.04) | |

Abbreviations: BC, blood culture; OR, odds ratio; CCI, Charlson Comorbidity Index; LRT, likelihood ratio test.

*Multivariable model (adjusted for age, sex, admission type, direct admission to ICU/HCU, LOS prior to blood culture test, and ward of BC test).

**P value for the interaction test was 0.17

been due to small sample size – there were relatively few patients who had BC performed without prior antibiotic treatment. After controlling for other factors, patients who had no antibiotic prior to BC test were likely to have positive BC at higher comorbidity score (aOR 1.16, 95%CI:0.54-2.49 for CCI score 1 and aOR 1.28, 95%CI: 0.27-2.49 for BC for CCI score >1) whereas patients who had antibiotic prior to BC test were less likely to have positive BC with rising score of comorbidities (aOR 0.84, 95%CI: 0.49-1.43 for CCI score 0, aOR 0.66, 95%CI: 0.38-1.16 for CCI score 1 and aOR 0.61, 95%CI: 0.36-1.04 for CCI score >1).

## Discussion

We investigated if the CCI, a widely known system of categorization of degree of medical comorbidity, could be used to identify patients at higher risk of positive BC in a cohort of patients admitted to one hospital in Indonesia. We found that a higher CCI score was not associated with BC positivity for any pathogens irrespective of the categorization of CCI scores, or the selection of pathogens.

This lack of association between comorbidity and positive BC could be because of prior antibiotic treatment. This was very common in our cohort and has been found before for elderly patients with multimorbidity – these patients frequently receive broad-spectrum antibiotics for long periods in this hospital and other tertiary care centers [12,13]. This would reduce the chance of obtaining a positive culture as obtaining BC during or after antibiotic therapy is associated with a reduction of pathogen detection [14].

Among patients with confirmed BSIs, a substantial proportion were due to antimicrobial-resistant (AMR) pathogens: 78% of *E. coli* were resistant to third-generation cephalosporin, 56% of *K. pneumoniae* were resistant to third-generation cephalosporin-resistant, 51% of *S. aureus* were resistant to methicillin, and 48% of *Acinetobacter* spp. were resistant to carbapenem [9]. The observed AMR patterns suggest that prior antibiotic exposure reduces the likelihood of detecting susceptible organism and culture positivity. Prior antibiotic use may reduce blood culture positivity, particularly when pathogens are susceptible to the administered agents. However, frequent or inappropriate antibiotic exposure can promote the emergence and selection of antimicrobial-resistant organisms, contributing to the high burden of AMR. This is further supported by the fact that we found an inverse relationship between continuous CCI and BC positivity for priority pathogens.

A cohort study exploring risk factors for Central Line Associated Blood Stream Infections (CLABSI) found that the composite CCI scores were not strongly associated with increased risk for CLABSI. However, the individual patient's comorbid conditions were strongly associated with either increased or decreased risk for CLABSI [15]. For example, patients with renal disease had 1.88 times the risk of CLABSI. In our study, the comorbid condition that had the highest positive BC proportion is severe renal disease as well, which indicates that possibly certain specific comorbid conditions are more relevant than others, and the scoring of the CCI may not be well adjusted to be used to predict BSI.

Focusing on five priority pathogens only, the presence of comorbidities slightly increased the odds of having a positive BC, particularly when considering the CCI score as a continuous variable. The priority pathogens, *S. aureus, E. coli, K. pneumoniae, P. aeruginosa and Acinetobacter* spp., are central to the threat of antimicrobial resistance [16]. Therefore, this finding suggests that patients with higher CCI score could be at higher risk of AMR BSI. The high levels of AMR in this institution are described more extensively in a prior analysis of the same dataset [9].

We found that there was non-significant effect modification by prior antibiotic treatment, most likely due to sparse data bias, as only 460/3,875 patients did not receive antibiotics prior to BC. However, the direction of the effect estimate indicated that in patients without prior antibiotic treatment and with a high CCI score (>1) had approximately 1.3 times the odds of having BC positive compared to patients without comorbidity. In comparison, patients with prior antibiotic treatment had less odds of having BC positive with more comorbidity. This suggests that the high utilisation rates of antibiotics in our setting may mask BSI, greatly reducing the utility of BC as a diagnostic tool. This highlights the urgent need for better implementation of diagnostic and antibiotic stewardship to take BC prior to antibiotic and reduce unnecessary antibiotic use.

In this hospital in Indonesia, we observed a relatively high BC contamination rate (9.9%) – which is likely caused by poor aseptic techniques during BC collection. This issue stems from the fact that most BC samples are collected by healthcare officers who lack specific training in this procedure, while trained laboratory staff, due to limited personnel, are only assigned to collect samples in the ICU. Following consultations with the hospital's infection prevention and control team and physicians, plans are underway to implement targeted interventions aimed at improving sample collection practices and reducing contamination rates.

The single most common pathogen causing BSI in this cohort was *Burkholderia cepacia* (240/3,875 patients). This organism has been identified in clinical samples in other countries in this region [17,18], but further work is needed to better understand its clinical and epidemiological significance.

The utility of the CCI may vary between hospital-acquired infections (HAIs) and community-acquired infections (CAIs). In HAIs, CCI may be less predictive of infection risk, as nosocomial factors such as invasive procedures, ICU admission, prolonged hospitalization, and exposure to antimicrobial-resistant pathogens are more influential in infection acquisition. In contrast, among patients with CAIs, the comorbidity burden captured by CCI may better reflect the individual's baseline vulnerability to infection and disease severity at presentation.

While our study has negative findings, it has several strengths. Our study utilized a relatively large single-centre dataset from a LMIC setting. A consistent approach was used to capture all hospitalized patient's data over four years and the amount of missing data is negligible. There may have been some information error due to inaccurate recording of ICD 10 codes, but we believe the effects of any resulting misclassification would be modest in this data set. Additionally, a systematic analytical approach was employed, including the use of directed acyclic graphs (DAGs) to identify and account for potential confounders, enhancing the validity of the results.

Our study also has limitations. Firstly, our analysis variables were constrained to routinely collected data in this hospital. Some relevant risk factors for BSI were therefore not possible to adjust for – for example, duration of intravenous catheterization, use of parenteral nutrition and data relating to inter-hospital transfer. A systematic review indicated these factors could all play a part in the risk of developing catheter-associated BSI [19]. This means that there may be residual confounding in this analysis. Secondly, the utilization of BC in this hospital was infrequent – a typical minimum recommendation is for approximately 100 BC/1,000 patient days [20], while our hospital reported 6.4 BC/1,000 patient days. The high positivity rate (20%) also implies that not enough BC were being taken, which means we might have missed data from patients with lower number of comorbidities or lower severity who may have a lower proportion of BC positivity. Comparison of characteristics of patients who had BC performed versus the whole inpatient population showed that clinicians were (understandably) more likely to perform BC in severely ill patients [21]. These characteristics of the dataset could have biased the results, but without substantially increased funding and support, hospitals in the LMICs cannot overcome this limitation - the central purpose of our analysis was to attempt to find better ways to target the limited resources available for performing BC.

In conclusion, in this single-hospital study utilizing routine available datasets, no significant association was observed between comorbidities as defined by the CCI score and positive BC results. This lack of association could be due to widespread antibiotic use before diagnostics, or lack of sensitivity of the CCI. To better guide BC as a diagnostic tool in LMICs settings, other factors will need to be considered in future research. In the meantime, improvement of antibiotic stewardship and training of personnel to take BC should be considered to improve BSI diagnosis and management in these settings.

## Supporting information

**S1 Data. CCI score and blood culture dataset.**
(XLS)

## Acknowledgments

We gratefully acknowledge the support provided by staff Dr. Wahidin Sudirohusodo Hospital, South Sulawesi, Indonesia.

## Author contributions

**Conceptualization:** Alexander M. Aiken.

**Data curation:** Patricia M. Tauran.

**Formal analysis:** Patricia M. Tauran.

**Funding acquisition:** Direk Limmathurotsakul.

**Investigation:** Patricia M. Tauran.

**Methodology:** Patricia M. Tauran, Alexander M. Aiken.

**Project administration:** Patricia M. Tauran.

**Resources:** Mansyur Arif.

**Supervision:** Mansyur Arif, Direk Limmathurotsakul, Marlieke E A de Kraker, Alexander M. Aiken.

**Validation:** Patricia M. Tauran, Alexander M. Aiken.

**Visualization:** Patricia M. Tauran, Alexander M. Aiken.

**Writing – original draft:** Patricia M. Tauran.

**Writing – review & editing:** Direk Limmathurotsakul, Marlieke E A de Kraker, Alexander M. Aiken.

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
