## [Decision Letter · Decision Letter 0]

PGPH-D-25-00198

Association between Charlson Comorbidity Index and positive blood cultures at a tertiary-care hospital in Indonesia

Dear Dr. Tauran,

Thank you for submitting your manuscript to PLOS Global Public Health. After careful consideration, we feel that it has merit but does not fully meet PLOS Global Public Health’s publication criteria as it currently stands. Therefore, we invite you to submit a revised version of the manuscript that addresses the points raised during the review process.

We look forward to receiving your revised manuscript.

Kind regards,

Sumanth Gandra

Academic Editor

Journal Requirements:

1.  Please provide additional details regarding participant consent. In the ethics statement in the Methods and online submission information, please ensure that you have specified (1) whether consent was informed and (2) what type you obtained (for instance, written or verbal, and if verbal, how it was documented and witnessed). If your study included minors, state whether you obtained consent from parents or guardians. If the need for consent was waived by the ethics committee, please include this information.

Additional Editor Comments (if provided):

The authors in this study examined the association between Charlson comorbidity index (CCI) and positive blood cultures among hospital acquired bacteremia cases. The reason for examining this association was to prioritize blood culture testing resources to high yield cases in resource limited settings. To examine their hypothesis, they utilized EMR data between 2015-2018 from a tertiary care hospital in Indonesia. The primary outcome was association between CCI and positive blood culture for any pathogen after adjusting potential confounders and prior antibiotic use as effect modifier. The strength of the study is large database (involving more than 61,000 cases with approximately 3900 patients with blood culture data available) with well thought data analysis. Overall, the manuscript is well written, but I have several methodological clarifications for authors:

1. Lines 118-123- Why were blood culture contaminants not excluded or adjusted for?

2. Lines 121 and 136- Authors need to explain by only IV antibiotics were included and not oral antibiotics before blood cultures were taken?

3. The other effect modifier would be the specific antibiotic patient was receiving when blood culture was positive. Considering the high antimicrobial resistance among pathogens recovered from positive blood cultures, it is possible that the antibiotic the patients received were selected for the resistant organisms or that the empiric antibiotic therapy was not appropriate. This was discussed in lines 292-293 but I wonder if this could be adjusted in the model.

4. Lines 327-330: B. cepacia is known to cause nosocomial outbreaks and this is an example of potential selection for resistant pathogen despite being on antibiotics. B.cepacia is intrinsically resistant to most beta-lactam antibiotics and therefore the above point I raised is important.

5. As authors state there are several issues studying the association of CCI with positive blood culture in an LMIC setting like the authors used- most of patients receive antibiotics prior to blood cultures, bias in selecting patients where blood cultures will be obtained, poor infection control practices, high blood culture contamination rates, nosocomial outbreaks. So, the question is whether this was even an appropriate setting to examine this question. Maybe, we should examine this association in high-income settings where the influence of the factors mentioned above is minimal.

Reviewers' comments:

Reviewer's Responses to Questions

**Comments to the Author**

1. Does this manuscript meet PLOS Global Public Health’s publication criteria?

Reviewer #1: Yes

2. Has the statistical analysis been performed appropriately and rigorously?

Reviewer #1: Yes

3. Have the authors made all data underlying the findings in their manuscript fully available (please refer to the Data Availability Statement at the start of the manuscript PDF file)?

Reviewer #1: No

4. Is the manuscript presented in an intelligible fashion and written in standard English?

Reviewer #1: Yes

Reviewer #1: This is a well-planned analysis looking at the association between Charlson Comorbidity Index and positive blood cultures. Detailed and well-written.

What is the rationale for choosing 2 days after admission as a cut-off for hospital acquired bacteremia? Is 2 days a long enough duration for someone to acquire a hospital infection?

Hospital-acquired infection is briefly mentioned in the abstract and introduction, but not really expanded in the discussion. Needs to be include to tie the manuscript together. How the use of CCI for hospital acquired infection different from its application in patient with community acquired infections who get hospitalized?

Abstract

- Is Low- and Middle-Income Countries, Intensive/High Care Unit all capital letters? – And throughout the rest of the manuscript, revise what needs to be capitalized.

- Line 38 – 40 include % if word count allows

- Line 43 – 45, the OR would be more informative than p-value alone.

- Improve conclusion of abstract – what other factors? Are these conclusions you can directly make from your results?

Introduction

- Line 52-53, reference?

- Once you have introduced the acronym, no need to include the full form? - line 71 and through methods and other sections for other acronyms as well.

Methods:

- Well described analysis methods.

- Although the methods are referenced, short description of this hierarchy rule would be helpful to understand scoring.

- Can you comment on the completeness and quality of the electronic records?

- What are progressive wards?

- What is the minimum and maximum CCI value you can get, rationale for the categorization?

Results

- Patient-days calculation not described in the methods – please include.

- Add percent for 3875 – line 171

- What was the median (range) blood cultures per participant?

- Line 178, add %

- Line 179 – 181 should be in the methods

- Line 190-191, 371/786=47.2%?

- Line 191, 192, add %

- Describe figures 3 and 4 any further? Figures need to be clear stand alone, add information as appropriate

- Line 208 – 214, what proportion of participants have more than one comorbidity?

- Minor comment, could make the tables cleaner by not having % in each row/col in the results.

Discussion:

- Line 288-294 not described in results. – clarify the paragraph. Not clear what the author means. How does it relate to results of the analysis?

- Line 295-297 Is not clear, does the author mean the CCI is not a risk factor for CLASBSI or was it the independent variable being studied?

**Do you want your identity to be public for this peer review?** For information about this choice, including consent withdrawal, please see our Privacy Policy

Reviewer #1: No

---

## [Editor Report · Decision Letter 1]

Association between Charlson Comorbidity Index and positive blood cultures at a tertiary-care hospital in Indonesia

PGPH-D-25-00198R1

Dear Dr. Tauran,

We are pleased to inform you that your manuscript 'Association between Charlson Comorbidity Index and positive blood cultures at a tertiary-care hospital in Indonesia' has been provisionally accepted for publication in PLOS Global Public Health.

Best regards,

Sumanth Gandra

Academic Editor